# Densification of Genetic Map and Stable Quantitative Trait Locus Analysis for Amino Acid Content of Seed in Soybean (*Glycine max* L.)

**DOI:** 10.3390/plants13152020

**Published:** 2024-07-23

**Authors:** Xi Li, Pingting Tan, Guoxi Xiong, Ronghan Ma, Weiran Gao, Aohua Jiang, Jiaqi Liu, Chengzhang Du, Jijun Zhang, Xiaochun Zhang, Li Zhang, Zelin Yi, Xiaomei Fang, Jian Zhang

**Affiliations:** 1College of Agronomy and Biotechnology, Southwest University, Chongqing 400715, China; 2Institute of Specialty Crop, Chongqing Academy of Agricultural Sciences, Chongqing 402160, China; 3Chongqing Three Gorges Academy of Agricultural Sciences, Chongqing 404100, China

**Keywords:** soybean, essential amino acid, QTL (quantitative trait locus), candidate gene

## Abstract

Soybean, a primary vegetable protein source, boasts favorable amino acid profiles; however, its composition still falls short of meeting human nutritional demands. The soybean amino acid content is a quantitative trait controlled by multiple genes. In this study, an F_2_ population of 186 individual plants derived from the cross between ChangJiangChun2 and JiYu166 served as the mapping population. Based on the previously published genetic map of our lab, we increased the density of the genetic map and constructed a new genetic map containing 518 SSR (simple sequence repeats) markers and 64 InDel (insertion-deletion) markers, with an average distance of 5.27 cm and a total length of 2881.2 cm. The content of eight essential amino acids was evaluated in the F_2:5_, F_2:6_, and BLUP (best linear unbiased prediction). A total of 52 QTLs (quantitative trait loci) were identified, and 13 QTL clusters were identified, among which loci02.1 and loci11.1 emerged as stable QTL clusters, exploring candidate genes within these regions. Through GO enrichment and gene annotation, 16 candidate genes associated with soybean essential amino acid content were predicted. This study would lay the foundation for elucidating the regulatory mechanisms of essential amino acid content and contribute to germplasm innovation in soybeans.

## 1. Introduction

Soybean (*Glycine max* L. Merr.) is one of the most important seed crops grown worldwide. Domestication and improvement have shaped soybean as the most important dual-function crop to prove highly valuable seed protein and oil, which together account for its high economic value [1,2]. Soybean is rich in protein, making it an essential plant-based protein source in human diets [3]. The bioavailability of soy protein in humans is equivalent to that of proteins derived from milk and eggs [4,5]. According to data published by the Food and Agriculture Organization (FAO), global soybean production in 2022 amounted to 348.8 million tons, of which only about 6% was used for direct human consumption, while approximately 75% was utilized as animal feed [6], thereby constituting a primary source of protein in animal feed formulations [7].

Soybean exhibits a relatively high protein content and it has an excellent amino acid composition [8]. However, the amino acid composition of soybean still presents certain deficiencies; for example, it is deficient in sulfur-containing amino acids [9,10], which is insufficient to meet the amino acid requirements of animals and humans [11,12]. It is a significant limiting factor for improving the seed quality of the lack of sulfur-containing amino acids in soybean [13]. Therefore, enhancing the content of certain amino acids can improve the nutritional value of soybeans [14]. Constructing a high-density genetic map and identifying relevant genes affecting QTLs of essential amino acid content can contribute to enhancing the nutritional value of soybean seed.

The content of essential amino acids is a complex quantitative trait governed by multiple genes and highly influenced by environmental factors, which poses challenges for plant breeders in selecting this trait [9,15,16]. QTL analysis offers a robust tool for soybean breeders, facilitating the discovery of novel sources of variation and the investigation of the genetic determinants underlying quantitatively inherited traits. According to the latest database, 112 QTLs related to essential amino acids have been localized in the SoyBase (http://www.soybase.org, accessed on 7 March 2024). Panthee et al. developed recombinant inbred lines using the N87-984-16, and TN93-99 crosses to identify genomic regions controlling amino acid content [17]. B. Fallen et al. constructed recombinant inbred lines using Essex and Williams82, identifying 10 QTLs related to amino acid content [18]. Wang et al. developed two populations and detected 8 QTLs associated with the content of methionine (Met) and lysine (Lys) [19]. Ma et al. used recombinant inbred lines from the Kefeng 1 and Nannong 1138-2. crosses to identify 9 QTLs associated with cysteine (Cys) and methionine (Met) [20].

This study aims to construct a high-density genetic map and identify QTLs related to essential amino acids in soybean seed. Utilizing a population derived from a cross between ChangJiangChun2 (CJC2) and JiYu166 (JY166) across three different environments, the findings are anticipated to aid marker-assisted selection (MAS) and improve our understanding of the genetic basis of essential amino acid composition in soybeans.

## 2. Results

### 2.1. Trait Phenotype Analysis

As shown in Table 1, it seems that the phenotype date of two parents appears to be different in eight traits in three environments. Among the eight traits, except for Trp and Phe, the levels of the other six amino acids in JY166 are relatively higher compared to those in CJC2. According to the phenotype data just mentioned, we can see apparent segregation within the population; the concrete information is as follows: coefficients of variation of different traits range from 0.44~22.2%, which shows transgressive segregation for each trait. The histogram of frequency distribution (Figure 1) showed a basically normal distribution of eight traits in the three environments, which was consistent with the genetic rule of quantitative traits.

Correlation analysis (Figure 2) showed that there was a certain correlation between eight essential amino acids. Trp demonstrated highly significant negative correlations with four amino acids and a notably strong negative correlation with Met. Lys had a positive correlation with Thr and Ile. The correlation coefficient between Phe and Leu was the largest, reaching 0.970, indicating that suitable varieties could be selected according to these laws in breeding.

### 2.2. Genetic Map Construction

Based on the resequencing data of CJC2 and JY166, 162 pairs of InDel primers were selected. After genotyping the parental lines, 64 pairs of primers were used in constructing the genetic map. Combined with the previous genetic map constructed in our laboratory [21], a new linkage map was constructed, which contained 582 maker loci distributed across the 20 chromosomes of soybeans. The genetic map was 2881.2 cm in length, with an average map distance of 5.27 cm (Figure 3). The longest linkage map was 217.4 cm of chromosome 18; the shortest was 60.2 cm of chromosome 16. The maximum number of markers was 63 on chromosome 19, and the minimum number of markers was 10 on chromosome 16. The longest average distance between markers was 9.27 cm on chromosome 8, and the shortest average distance between markers was 2.43 on chromosome 19.

### 2.3. QTLs Identified for Essential Amino Acids

Using the multiple QTL model (MQM) mapping methods and based on the constructed linkage groups, a total of 52 stable QTLs associated with essential amino acids were mapped in more than two environments. (Figure 4 and Table 2).

A total of 10 QTLs for valine were identified and mapped on seven chromosomes, explaining the phenotypic variation from 8.40% to 20.30%. The favorable alleles of three QTLs were derived from CJC2, and the favorable alleles of the other three QTLs were derived from JY166.

A total of 5 QTLs for threonine were identified and mapped on five chromosomes, explaining the phenotypic variation from 9.70% to 13.0%. Most of the QTL’s favorable alleles were derived from CJC2, except for qThr18.1.

A total of 13 QTLs for phenylalanine were identified and mapped on ten chromosomes, explaining the phenotypic variation from 8.10% to 23.80%. The favorable alleles of ten QTLs were derived from CJC2, and the favorable alleles of the other three QTLs were derived from JY166.

A total of 4 QTLs for methionine were identified and mapped on four chromosomes, explaining the phenotypic variation from 8.20% to 13.60%. All favorable alleles were derived from JY166.

A total of 6 QTLs for Lysine were identified and mapped on six chromosomes, explaining the phenotypic variation from 7.40% to 17.20%. The favorable alleles of five QTLs were derived from CJC2, and the favorable alleles of one other QTL were derived from JY166.

A total of 7 QTLs for Leucine were identified and mapped on six chromosomes, explaining the phenotypic variation from 7.40% to 17.20%. The favorable alleles of the QTLs were derived from CJC2, except for qLeu11.1.

A total of 5 QTLs for Isoleucine were identified and mapped on 4 chromosomes, explaining the phenotypic variation from 8.90% to 14.60%. All favorable alleles of the QTLs were derived from CJC2.

A total of 2 QTLs for Tryptophan were identified and mapped on 2 chromosomes, explaining the phenotypic variation of 7.40% and 17.20%. The favorable alleles of the QTLs were derived from JY166.

### 2.4. Identification and Analysis of QTL Clusters

Following the principle of stability and effectiveness, a total of 13 QTL clusters were located on 10 chromosomes in this study (Table 3). In terms of controlling quantitative traits, Loci02.1 contains the highest number of a total of 7 QTLs. One QTL cluster of five traits was Loci12.1, and two QTL clusters of four traits were Loci12.2 and Loci17.1, while the one QTL cluster of three traits was Loci19.1. The remaining QTL clusters were all for two traits. In terms of the number of controlled traits and stability of detected QTL, two important clusters were Loci02.1 and Loci11.1.

### 2.5. Candidate Gene Prediction

In the respective promising intervals of the two important clusters, the Loci02.1 searched 236 genes in the physical location ranging from 43.68 MB to 45.65 MB of chromosome 2, and the Loci11.1 searched 203 genes in location ranging from 10.12 MB to 12.78 MB of chromosome 11. Based on the GO enrichment tools of the SoyBase (http://www.soybase.org, accessed on 9 March 2024) and the Wm82 genome assemblies, all the genes were conducted with GO analysis (Figure 5). Among the genes of Loci02.1 and Loci11.1, 34 genes and 31 genes, respectively, failed to be found in any GO Ontologies. By integrating gene functional annotation, a total of 16 candidate genes potentially involved in regulating the essential amino acid content were identified (Table 4).

## 3. Discussion

Soybean is a crucial crop globally, boasting high protein content and an excellent amino acid composition. However, challenges persist in the seed essential amino acid content of soybeans, as it is deficient in sulfur-containing amino acids. Previous research indicates that simply increasing soybean crude protein content may not elevate essential amino acid concentration [22]. Hence, it is imperative to investigate QTL associated with soybean essential amino acid content.

The limit of map-based cloning in soybeans includes the insufficiency of molecular markers [23]. In this study, QTL mapping was performed on the F_2_ population obtained by hybridization of CJC2 and JY166, and a genetic map containing 564 SSR markers and 64 InDel markers was constructed, with an average map distance of 5.27 cm (Figure 3). Compared with previous studies on QTL mapping of essential amino acid content in soybeans, the map has a higher marker density, which improves the QTL resolution and helps to fine-locate candidate genes [17,22,24].

In this study, the content of eight essential amino acids showed extensive and continuous variation, and there was clear transgressive segregation (Figure 1), indicating that these traits are complex quantitative traits controlled by multiple genes, which is consistent with previous results [18,22]. A total of 52 stable QTLs were detected by MQM, and it was observed that the most favorable alleles came from CJC2. Of these QTLs, qIle01.1 was consistent with Panthee et al., and qPhe19.1 had a high LOD value (8.97) and phenotypic variance (23.8%); this region may contain candidate genes controlling Phe (Table 2).

We detected overlapping QTLs for multiple traits and 13 QTL clusters located on chromosomes 1, 2, 3, 6, 7, 11, 12, 17, 18, and 19 (Table 3). Each QTL cluster was associated with two or more traits related to seed essential amino acids. In terms of the number of controlled traits and QTL environmental stability, Loci02.1 and Loci11.1 might be selected for further research. QTL clusters may represent gene/QTL linkage or pleiotropic effects of a single QTL within the same genomic region [25]. The correlation analysis shows that the correlation between Phe and Leu is the largest, reaching 0.970, and 6 QTL clusters associated with Phe and Leu, which deserves further consideration. Fallen et al. also reported a positive correlation between these two amino acids [18]. The physical locations of Loci02.1 and Loci11.1 range from 43.68 MB to 45.65 MB and from 10.12 MB to 12.78 MB, respectively, and 439 genes were obtained within the intervals. Eventually, after gene function annotation screening, 16 candidate genes for seed essential amino acid of soybean are obtained.

Among these candidate genes, we identified several with homologs in Arabidopsis thaliana (AT), some of which are associated with our target traits. Here, I present them to provide a reference for further investigation (Table 4). *Glyma.02g254300* encodes a protein containing Leu-rich repeats and a degenerate F-box motif and is related to the genetic pathway that modulates petal senescence by jasmonic acid (JA) [26]. *Glyma.02g260900* was found to be related to the synthesis of Tyrosine and typically strongly feedback inhibited by Tyr [27]. *Glyma.02g263900* was related to the male and female gametes in the sexual reproduction [28]. *Glyma.02g270000* could be related to the function of regulating the hexameric structure and ATPase activity of AtCDC48 [29]. *Glyma.02g270700* encoded the leucine-rich repeat-malectin receptor kinases for Arabidopsis immune responses triggered by β-1,4-D-Xylo-oligosaccharides from plant cell walls [30]. *Glyma.02g271700* was important in chromatin regulation and in maintaining transcriptional gene silencing (TGS) in some genomic regions of AT [31]. *Glyma.11g144900* could be involved in many development processes, including the regulation of premature cell death [32].

In light of the relevant literature in the field, the 16 genes identified are deemed crucial candidate regulators of soybean essential amino acid content, as determined by QTL localization and gene function annotation. Nonetheless, their precise functional mechanisms necessitate further scrutiny.

## 4. Materials and Methods

### 4.1. Plant Materials

Changjiang Chun 2 (CJC2) is a high-protein variety from Chongqing, while JY166 is a widely cultivated high-oil variety in northern regions. There is a significant genetic difference between parental lines, and they have a distant genetic relationship. In this study, 186 individual plants of the F_2_ population produced from the hybridization of CJC2 and JY166 were used as the location population.

The F_2:5_ population was planted during the summer of 2023 at the Teaching and Experimental base of Southwest University in Chongqing (23CQ), while the F_2:6_ population was planted in the winter of 2021 in Yuanjiang, Yunnan (23YN). Both populations were sown in single rows, with a row length of 1 m, row width of 0.5 m, and plot spacing of 0.2 m, with two seeds planted per plot. Following standard field management practices, all plants were harvested at maturity, and subsequent testing was conducted to determine the essential amino acid content of the seed.

### 4.2. DNA Extraction and Genotyping

Genomic DNA was extracted from 189 plants, including the F_2_ population, the two parents, and F_1_ plant. A total of 2933 SSR primer pairs were synthesized by Biotech Bioengineering Co., Ltd. (Shanghai, China), derived from the soybean database SoyBase (http://www.soybase.org, accessed on 9 March 2024) [33]. Some of these BARCSOYSSR primers were renamed as SWU in this study. And a total of 162 insertion-deletion (InDel) primer pairs were synthesized based on comparing parental 10 × resequencing data with the soybean Wm82.a4.v1 reference genome, selecting InDel sites > 15 bp as markers. The primer sequence of SSR and InDel were listed in Appendix A. The PCR amplification, following the protocol described by Zhang et al. [34], utilized primers containing polymorphisms between the two mapping parents for genotyping the individual plants of the F_2_ population. The band types identical to CJC2 and JY166 were recorded as A and B, respectively, while the heterozygous band type was labeled as H, and the deletion was denoted as U. The resulting data were then collected for further analysis, revealing the definition of additive effects for the CJC2 allele, signifying that positive genetic effects were present, indicating that CJC2 alleles contribute to increased phenotypic values.

### 4.3. Determination of Traits

FOSS NIRS DS2500 (Foss Analyical A/S, Hilleroed, Denmark) was used to determine 8 essential amino acid content of seeds, from 400 to 2500, in transmittance mode with a 1 mm pathlength. A reference scan was taken once in every 10 sample scans. To increase the signal-to-noise ratio, both reference and sample spectra were averaged from 32 scans. Samples were temperature equilibrated at 33 °C (approximately 3 min) in the instrument before scanning and for the rest.

The phenotype data were subjected to statistical analysis using Excel 2019, while the data were processed and plotted using Origin 2019. The best linear unbiased prediction (BLUP) values were compute based on the amino acid content of F_2:5_ and F_2:6_ using R programming language [35].

### 4.4. Map Construction and QTL Detection

Conduct marker linkage analysis using JoinMap4.0 to establish a genetic linkage map with an LOD score of 3.0 and employ the Kosambi mapping function for mapping unit conversion [36,37]. QTL localization for all traits was analyzed with the MQM and MapQTL6.0 software, and phenotypic data were analyzed using 1000 permutation tests with significance *p* = 0.05 and LOD = 3.0 as the threshold to determine the presence of QTLs. The QTL graphic representation of the linkage groups was created using MapChart2.2 [38]. The qualified interval was then named QTL. The QTLs were named with the letter “q”, the trait name, the chromosome number, and the sort number. For example, the first QTL we found at Chromosome 1 related to Val would be called qVal01.1.

### 4.5. QTL Cluster Identification

A QTL cluster is a densely populated QTL region of the chromosome which contains multiple QTLs associated with various traits [39]. All QTLs will be sorted based on their physical positions on their respective chromosomes. If there are two or more QTLs at the same physical position, they will be grouped into a QTL cluster. The QTL clusters that we found were labeled with “Loci”. For example, for the QTL cluster denoted as Loci01.1, Loci indicates a QTL cluster, 01 indicates the chromosome on which the QTL cluster was detected, and 01.1 indicates the order of the QTL cluster identified on the chromosome.

### 4.6. Candidate Gene Prediction

The candidate genes were searched with SoyBase (http://www.soybase.org, accessed on 10 March 2024) on the candidate interval of promising QTL clusters. Enriched the terms GO (Gene Ontology) and analyzed the families and subfamilies, molecular functions, biological processes, and pathways of genes in the identified QTLs. Finally, candidate genes related to essential amino acids were screened.

## 5. Conclusions

In this study, the genetic map previously constructed in the laboratory was densified by adding 64 InDel markers. Using the MQM method, QTLs associated with essential amino acids in soybeans were mapped, identifying 52 stable QTLs. By integrating Gene Ontology (GO) enrichment analysis and gene function annotation, 16 genes related to the target traits were ultimately identified. The candidate genes delineated in this study furnish pivotal theoretical underpinnings and genetic reservoirs for the subsequent enhancement of soybean essential amino acid content.

## Figures and Tables

**Figure 1 plants-13-02020-f001:**
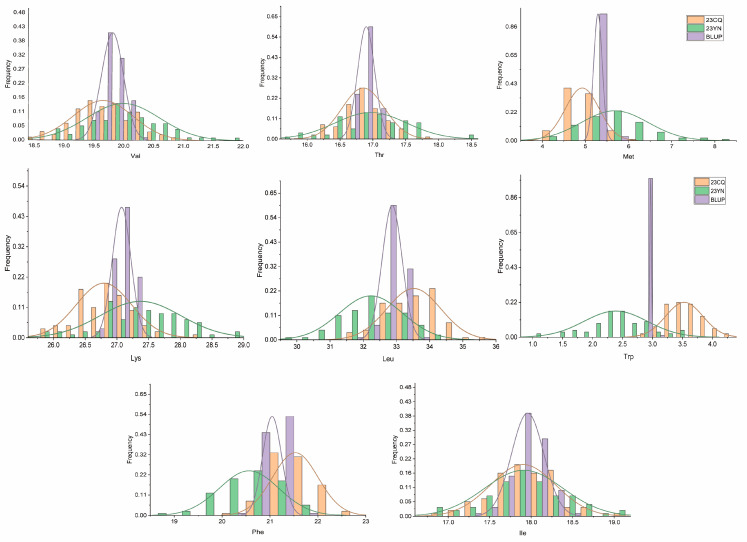
Distribution of eight essential amino acid content in F_2:5_, F_2:6_, and BLUP.

**Figure 2 plants-13-02020-f002:**
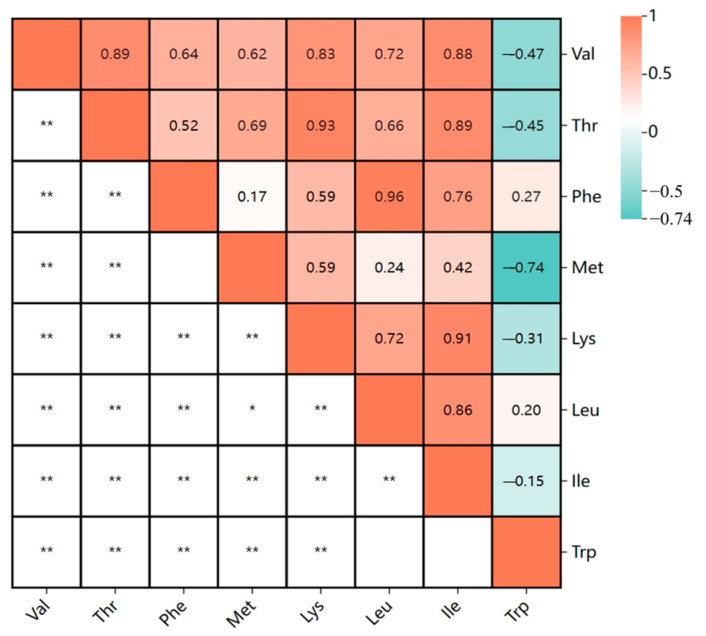
Correlation analysis of eight essential amino acid contents (** = *p* < 0.001, * = *p* < 0.05. The data in this table are the average results of three environments).

**Figure 3 plants-13-02020-f003:**
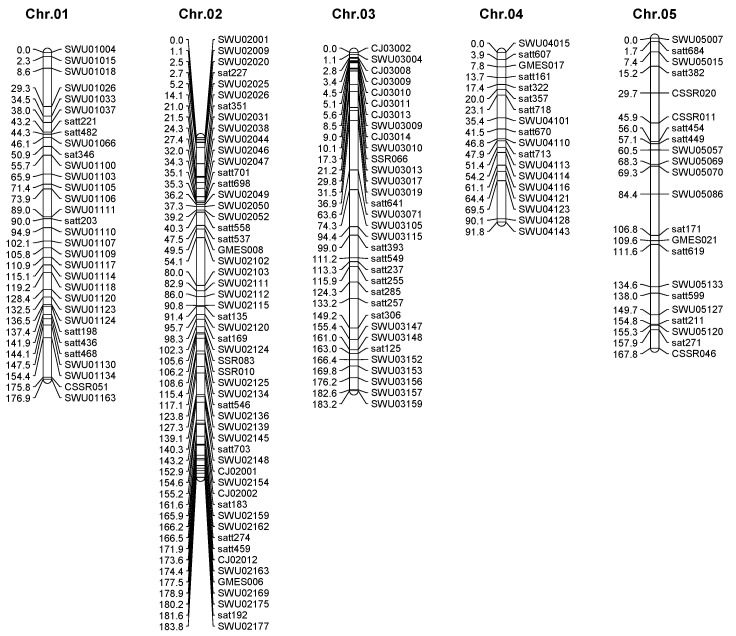
Linkage map derived from (CJC2 × JY166) F_2_ population.

**Figure 4 plants-13-02020-f004:**
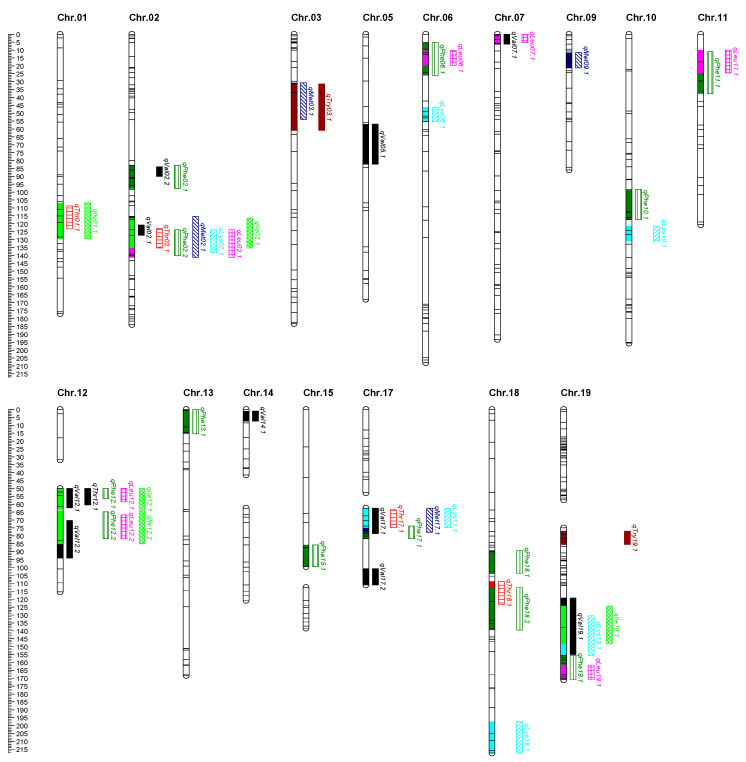
QTLs for essential amino acids derived from (CJC2 × JY166) population.

**Figure 5 plants-13-02020-f005:**
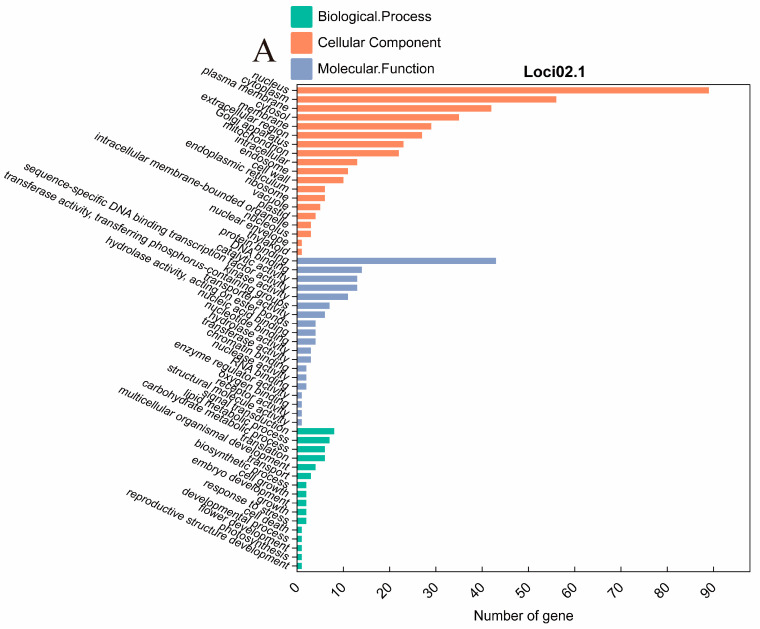
GO term enrichment analysis of the genes located within the two QTL clusters: (**A**) Loci02.1 and (**B**) Loci11.1.

**Table 1 plants-13-02020-t001:** Characteristics of seed essential amino acid contents in three environments.

Traits	Env.	Parent		Popuation							
		CJC2	JY166	Max.	Min.	Mean	S.D	Variance	CV/%	Kurt	Skew
Valine (Val)	23CQ	19.40	18.40	20.90	18.50	19.65	0.50	0.25	2.55	−0.23	0.01
	23YN	18.20	19.97	21.80	18.80	19.99	0.61	0.37	3.05	0.18	0.34
	BLUP	19.36	19.54	20.30	19.41	19.81	0.18	0.03	0.93	−0.36	0.18
Threonine (Thr)	23CQ	16.90	16.00	17.90	16.20	16.85	0.32	0.10	1.91	0.61	0.14
	23YN	15.50	16.94	18.40	15.70	16.97	0.54	0.30	3.21	0.23	0.01
	BLUP	16.62	16.72	17.19	16.62	16.90	0.13	0.02	0.77	−0.31	0.08
Phenylalanine (Phe)	23CQ	22.10	21.00	22.70	20.40	21.53	0.47	0.22	2.19	0.08	−0.02
	23YN	19.60	20.56	21.80	18.90	20.56	0.62	0.38	3.02	−0.18	−0.14
	BLUP	20.95	20.93	21.60	20.40	21.04	0.20	0.04	0.94	0.94	−0.17
Methionine (Met)	23CQ	4.50	4.20	6.00	4.00	4.93	0.39	0.15	7.95	0.06	0.24
	23YN	5.00	5.65	8.00	4.00	5.68	0.78	0.61	13.72	0.56	0.53
	BLUP	5.16	5.20	5.63	5.02	5.29	0.10	0.01	1.89	1.14	0.35
Lysine (Lys)	23CQ	27.30	25.80	28.20	25.80	26.80	0.43	0.19	1.62	0.50	0.15
	23YN	25.70	27.34	28.90	25.90	27.37	0.64	0.41	2.33	0.04	−0.15
	BLUP	26.87	26.88	27.39	26.67	27.07	0.15	0.02	0.54	−0.12	−0.13
Leucine (Leu)	23CQ	34.20	32.40	35.70	31.50	33.52	0.81	0.65	2.41	−0.06	−0.16
	23YN	30.10	32.21	34.30	29.60	32.23	0.95	0.91	2.96	−0.26	−0.14
	BLUP	32.55	32.62	33.69	31.91	32.87	0.30	0.09	0.91	0.75	−0.16
Isoleucine (Ile)	23CQ	18.20	17.20	19.10	16.90	17.90	0.39	0.15	2.17	0.53	−0.02
	23YN	16.40	17.89	19.10	16.80	17.91	0.47	0.22	2.63	0.22	0.01
	BLUP	18.04	17.65	18.41	17.32	17.96	0.19	0.04	1.08	0.51	−0.27
Tryptophan (Trp)	23CQ	4.20	4.10	4.20	2.90	3.53	0.31	0.09	8.70	−0.63	0.08
	23YN	2.90	2.40	3.50	1.00	2.39	0.53	0.28	22.22	0.41	−0.20
	BLUP	3.00	2.98	3.00	2.94	2.97	0.01	0.00	0.44	0.09	0.13

23CQ and 23YN indicate the summer of 2023 in Chongqing and the winter of 2023 in Yunnan, respectively. Best linear unbiased prediction (BLUP) obtained by calculating the essential amino acid content of F_2:5_ and F_2:6_.

**Table 2 plants-13-02020-t002:** QTLs identified for essential amino acids in three environments.

QTL	Env. ^a^	Chr.	Nearest Marker	Interval (cm)	LOD	PVE (%) ^b^	Additive	Dominance
qVal02.1	23CQ	2	SWU02139	120.80–127.32	3.26	8.40	0.16	−0.10
	BLUP	2	SWU02139	123.78–133.32	4.81	11.30	0.06	−0.07
qVal02.2	23YN	2	SWU02111	83.91–87.96	3.44	9.90	0.23	0.10
	BLUP	2	SWU02111	83.91–89.96	3.51	8.40	0.06	0.02
qVal05.1	23YN	5	SWU05070	60.09–81.33	4.79	13.50	0.19	0.34
	BLUP	5	SWU05070	57.02–82.33	5.68	13.30	0.07	0.09
qVal07.1	23CQ	7	SWU07003	0.00–4.32	3.93	10.10	−0.01	0.33
	BLUP	7	SWU07003	0.00–6.32	3.21	12.70	−0.01	0.13
qVal12.1	23YN	12	SWU12060	0.00–12.31	4.76	13.40	0.19	0.32
	BLUP	12	SWU12060	0.00–12.31	5.67	13.20	0.05	0.11
qVal12.2	23CQ	12	satt302	20.31–44.01	5.18	13.10	0.19	−0.32
	BLUP	12	satt302	20.31–44.01	4.38	8.40	0.04	0.10
qVal14.1	23YN	14	SWU14003	1.00–7.38	4.74	13.40	0.25	0.12
	BLUP	14	SWU14003	1.00–6.38	3.92	20.30	0.11	−0.01
qVal17.1	23CQ	17	satt672	0.00–10.02	4.85	11.60	0.22	−0.03
	BLUP	17	satt672	0.00–16.06	9.09	9.80	0.03	0.11
qVal17.2	23YN	17	satt543	39.79–48.35	4.68	13.20	0.07	0.45
	BLUP	17	satt543	38.17–46.35	3.52	8.60	−0.05	0.06
qVal19.1	23YN	19	satt229	44.29–77.98	4.64	13.10	0.25	0.27
	BLUP	19	satt229	46.29–79.98	5.25	12.30	0.09	0.04
qThr01.1	23CQ	1	SWU01114	108.81–123.18	4.30	11.00	0.10	−0.14
	23YN	1	SWU01114	113.86–116.14	3.49	10.00	0.05	0.41
qThr02.1	23YN	2	SWU02139	123.78–132.32	4.70	13.30	0.08	−0.36
	BLUP	2	SWU02139	123.08–135.32	5.53	12.90	0.03	−0.08
qThr12.1	23YN	12	SWU12060	0.00–9.48	3.83	10.90	0.19	0.19
	BLUP	12	satt279	2.21–10.48	4.09	9.70	0.04	0.04
qThr17.1	23YN	17	CSSR036	2.00–12.27	4.10	11.70	0.25	0.06
	BLUP	17	satt672	1.00–12.27	4.83	11.40	0.06	0.00
qThr18.1	23YN	18	GMES144	212.53–217.37	4.69	13.20	−0.13	0.30
	BLUP	18	GMES107	206.95–215.83	4.41	10.50	−0.01	0.09
qPhe02.1	23YN	2	SWU02120	82.91–97.74	5.06	14.20	0.29	0.18
	BLUP	2	SWU02120	91.38–97.74	4.50	10.60	0.08	0.05
qPhe02.2	23YN	2	SWU02139	123.78–140.25	6.25	17.90	0.37	−0.01
	BLUP	2	SWU02139	123.78–140.25	7.47	17.10	0.10	−0.04
qPhe06.1	23YN	6	CJ06013	12.84–24.14	4.12	11.70	0.31	0.09
	BLUP	6	CJ06012	5.27–26.14	4.40	10.40	0.09	0.01
qPhe10.1	23CQ	10	GMES055	98.16–114.13	3.15	8.20	0.17	0.13
	23YN	10	satt473	110.16–115.92	3.56	10.20	0.23	0.22
	BLUP	10	GMES055	101.16–117.49	5.02	11.80	0.08	0.06
qPhe11.1	23CQ	11	GMES069	11.00–24.47	3.61	9.30	0.19	−0.07
	23YN	11	GMES069	12.00–29.60	4.39	12.40	0.29	−0.06
	BLUP	11	GMES069	7.00–37.60	6.37	14.70	0.10	−0.02
qPhe12.1	23CQ	12	satt279	0.00–6.48	5.55	14.00	0.07	0.36
	BLUP	12	SWU12060	0.00–3.21	4.78	11.30	0.03	0.13
qPhe12.2	23CQ	12	SWU12031	24.93–31.91	4.29	11.00	0.18	0.12
	23YN	12	SWU12031	16.81–31.91	5.86	16.30	0.28	0.24
	BLUP	12	SWU12031	14.81–31.91	7.64	17.40	0.09	0.08
qPhe13.1	23YN	13	sat039	0.00–13.03	3.58	10.30	0.21	−0.35
	BLUP	13	satt149	0.00–15.39	3.85	9.20	0.07	−0.10
qPhe15.1	23YN	15	satt573	85.70–99.41	4.00	11.40	−0.35	−0.12
	BLUP	15	CSSR014	90.29–99.41	4.38	10.40	−0.11	−0.09
qPhe17.1	23YN	17	sat220	12.27–19.39	7.29	19.80	0.40	0.03
	BLUP	17	sat220	11.27–19.39	6.90	15.90	0.11	0.00
qPhe18.1	23YN	18	SWU18055	89.50–103.74	4.02	11.50	−0.20	−0.30
	BLUP	18	SWU18055	90.09–100.92	3.39	8.10	−0.03	−0.12
qPhe18.2	23YN	18	sat358	112.44–135.51	6.29	9.60	−0.14	0.32
	BLUP	18	SWU18062	121.51–139.54	4.24	10.10	−0.06	−0.08
qPhe19.1	23YN	19	satt373	86.71–95.95	8.97	23.80	0.46	0.05
	BLUP	19	satt373	80.64–95.95	6.53	15.10	0.12	0.00
qMet02.1	23CQ	2	SWU02139	115.36–141.25	5.40	13.60	0.19	−0.03
	BLUP	2	SWU02136	116.36–125.78	3.42	8.20	0.02	−0.04
qMet03.1	23CQ	3	satt641	30.76–45.90	5.22	13.20	0.11	0.20
	BLUP	3	satt641	35.51–53.90	4.30	10.20	0.03	0.04
qMet09.1	23CQ	9	SWU09056	11.57–19.03	4.71	12.00	0.09	−0.23
	23YN	9	SWU09060	17.03–21.43	4.02	11.50	0.15	0.54
qMet17.1	BLUP	17	GMES100	0.00–15.27	4.08	9.70	0.04	0.02
	23CQ	17	sat326	0.00–5.85	3.93	10.10	0.16	−0.03
qLys02.1	23CQ	2	SWU02145	134.32–137.32	3.14	8.10	0.11	−0.22
	23YN	2	SWU02139	123.78–134.32	5.93	16.40	0.20	−0.36
	BLUP	2	SWU02139	123.78–138.32	6.21	14.40	0.05	−0.09
qLys06.1	23CQ	6	SWU06041	46.31–55.40	5.19	13.10	0.19	0.07
	BLUP	6	SWU06041	48.31–53.01	3.49	8.40	0.05	0.03
qLys10.1	BLUP	10	SWU10052	121.49–129.91	4.13	9.80	0.05	0.03
	23YN	10	SWU10052	123.49–130.91	3.92	11.20	0.19	0.25
qLys17.1	23YN	17	satt672	0.00–16.06	5.51	15.40	0.34	0.06
	BLUP	17	satt672	1.00–12.27	4.73	11.20	0.06	−0.01
qLys18.1	23YN	18	GMES107	197.52–212.53	6.25	17.20	−0.09	0.54
	BLUP	18	GMES107	207.95–217.37	4.19	10.00	0.00	0.10
qLys19.1	23YN	19	satt229	55.29–80.64	3.91	11.20	0.28	0.11
	BLUP	19	satt229	57.29–64.98	3.06	7.40	0.05	0.01
qLeu02.1	23YN	2	SWU02139	123.78–141.25	7.23	19.70	0.52	−0.14
	BLUP	2	SWU02139	123.78–138.32	5.90	13.70	0.12	−0.08
qLeu06.1	23YN	6	CJ06012	10.30–19.84	3.83	11.00	0.45	0.14
	BLUP	6	CJ06013	10.30–16.84	3.17	7.60	0.12	0.02
qLeu07.1	23CQ	7	SWU07003	0.00–5.32	3.79	9.70	0.14	0.48
	BLUP	7	SWU07001	0.00–5.32	4.73	11.20	0.03	0.20
qLeu11.1	BLUP	11	GMES069	10.00–24.47	4.52	10.70	0.13	−0.02
	23YN	11	GMES069	13.00–21.47	3.90	11.10	0.42	−0.03
qLeu12.1	23CQ	12	satt279	0.00–7.48	4.97	12.60	0.22	0.51
	BLUP	12	SWU12060	0.00–8.48	6.25	14.50	0.08	0.19
qLeu12.2	23YN	12	SWU12031	16.81–31.91	6.43	17.70	0.44	0.43
	BLUP	12	SWU12031	20.93–31.91	6.16	14.30	0.12	0.13
qLeu19.1	23YN	19	satt373	86.71–95.95	7.95	21.40	0.67	0.13
	BLUP	19	satt373	87.71–94.73	4.49	10.60	0.15	−0.01
qIle01.1	23CQ	1	SWU01114	106.81–129.4	5.83	14.60	0.14	−0.17
	23YN	1	SWU01117	108.81–113.86	3.84	11.00	0.04	0.43
qIle02.1	23YN	2	SWU02139	116.36–132.32	5.09	14.30	0.15	−0.23
	23CQ	2	SWU02139	128.32–135.32	3.45	8.90	0.13	−0.16
qIle12.1	23YN	12	SWU12060	0.00–3.21	4.00	11.40	0.15	0.21
	BLUP	12	SWU12060	0.00–13.31	5.02	11.90	0.05	0.11
qIle12.2	23CQ	12	satt302	14.31–35.07	4.22	10.80	0.18	−0.17
	BLUP	12	SWU12031	23.93–31.91	6.04	14.20	0.08	0.09
qIle19.1	BLUP	19	satt229	49.29–72.98	4.01	9.60	0.08	0.04
	23YN	19	satt229	60.29–72.98	4.08	11.60	0.20	0.18
qTrp03.1	23CQ	3	satt641	31.51–43.90	3.57	9.20	−0.09	−0.11
	BLUP	3	satt641	36.90–60.90	4.28	10.20	0.00	−0.01
qTrp19.1	23YN	19	satt156	2.00–8.63	4.68	13.20	−0.21	−0.32
	BLUP	19	sat340	4.00–10.63	4.08	9.70	−0.01	−0.01

^a^ 23CQ and 23YN indicate 2023 in Chongqing and 2023 winter in Yunnan, respectively. ^b^ PVE: phenotypic variance explained.

**Table 3 plants-13-02020-t003:** QTL clusters associated with essential amino acids in soybean.

Cluster	Chr.	Contained QTL	Env. ^a^	Interval (cm)	LOD
Loci01.1	1	qThr01.1	23CQ	108.81–123.18	4.3
			23YN	113.86–116.14	3.49
		qIle01.1	23CQ	106.81–129.4	5.83
			23YN	108.81–113.86	3.84
Loci02.1	2	qVal02.1	23CQ	120.80–127.32	3.26
			BLUP	123.78–133.32	4.81
		qThr02.1	23YN	123.78–132.32	4.7
			BLUP	123.08–135.32	5.53
		qPhe02.2	23YN	123.78–140.25	6.25
			BLUP	123.78–140.25	7.47
		qMet02.1	23CQ	115.36–141.25	5.4
			BLUP	116.36–125.78	3.42
		qLys02.1	23CQ	134.32–137.32	3.14
			23YN	123.78–134.32	5.93
			BLUP	123.78–138.32	6.21
		qLeu02.1	23YN	123.78–141.25	7.23
			BLUP	123.78–138.32	5.9
		qIle02.1	23YN	116.36–132.32	5.09
			23CQ	128.32–135.32	3.45
Loci02.2	2	qVal02.2	23YN	83.91–87.96	3.44
			BLUP	83.91–89.96	3.51
		qPhe02.1	23YN	82.91–97.74	5.06
			BLUP	91.38–97.74	4.5
Loci03.1	3	qMet03.1	23CQ	30.76–45.90	5.22
			BLUP	35.51–53.90	4.3
		qTrP03.1	23CQ	31.51–43.90	3.57
			BLUP	36.90–60.90	4.28
Loci06.1	6	qPhe06.1	23YN	12.84–24.14	4.12
			BLUP	5.27–26.14	4.4
		qLeu06.1	23YN	10.30–19.84	3.83
			BLUP	10.30–16.84	3.17
Loci07.1	7	qVal07.1	23CQ	0.00–4.32	3.93
			BLUP	0.00–6.32	3.21
		qLeu07.1	23CQ	0.00–5.32	3.79
			BLUP	0.00–5.32	4.73
Loci011.1	11	qPhe11.1	23CQ	11.00–24.47	3.61
			23YN	12.00–29.60	4.39
			BLUP	7.00–37.60	6.37
		qLeu11.1	BLUP	10.00–24.47	4.52
			23YN	13.00–21.47	3.9
Loci012.1	12	qVal12.1	23YN	0.00–12.31	4.76
			BLUP	0.00–12.31	5.67
		qThr12.1	23YN	0.00–9.48	3.83
			BLUP	2.21–10.48	4.09
		qPhe12.1	23CQ	0.00–6.48	5.55
			BLUP	0.00–3.21	4.78
		qLeu12.1	23CQ	0.00–7.48	4.97
			BLUP	0.00–8.48	6.25
		qIle12.1	23YN	0.00–3.21	4
			BLUP	0.00–13.31	5.02
Loci012.2	12	qVal12.2	23CQ	20.31–44.01	5.18
			BLUP	20.31–44.01	4.38
		qPhe12.2	23CQ	24.93–31.91	4.29
			23YN	16.81–31.91	5.86
			BLUP	14.81–31.91	7.64
		qLeu12.2	23YN	16.81–31.91	6.43
			BLUP	20.93–31.91	6.16
		qIle12.2	23CQ	14.31–35.07	4.22
			BLUP	23.93–31.91	6.04
Loci017.1	17	qVal17.1	23CQ	0.00–10.02	4.85
			BLUP	0.00–16.06	9.09
		qThr17.1	23YN	2.00–12.27	4.1
			BLUP	1.00–12.27	4.83
		qMet17.1	BLUP	0.00–15.27	4.08
			23CQ	0.00–5.85	3.93
		qLys17.1	23YN	0.00–16.06	5.51
			BLUP	1.00–12.27	4.73
Loci018.1	18	qThr18.1	23YN	212.53–217.37	4.69
			BLUP	206.95–215.83	4.41
		qLys18.1	23YN	197.52–212.53	6.25
			BLUP	207.95–217.37	4.19
Loci019.1	19	qVal19.1	23YN	44.29–77.98	4.64
			BLUP	46.29–79.98	5.25
		qLys19.1	23YN	55.29–80.64	3.91
			BLUP	57.29–64.98	3.06
		qIle19.1	BLUP	49.29–72.98	4.01
			23YN	60.29–72.98	4.08
Loci019.2	19	qPhe19.1	23YN	86.71–95.95	8.97
			BLUP	80.64–95.95	6.53
		qLeu19.1	23YN	86.71–95.95	7.95
			BLUP	87.71–94.73	4.49

^a^ 23CQ and 23YN indicate 2023 in Chongqing, and 2023 winter in Yunnan, respectively.

**Table 4 plants-13-02020-t004:** Candidate genes for essential amino acid of soybean.

Gene	Arabidopsis Homologs	GO ID	Gene Functional Annotation
*Glyma.02g250400*	*AT2G39750*	GO:0005634	nucleus
*Glyma.02g254000*	*AT1G36280*	GO:0003824	catalytic activity
*Glyma.02g254300*	*AT2G39940*	GO:0005515	protein binding
*Glyma.02g256100*	*AT3G47570*	GO:0016301	kinase activity
*Glyma.02g256200*	*AT3G47570*	GO:0005515	protein binding
*Glyma.02g260900*	*AT5G34930*	GO:0000166	nucleotide binding
*Glyma.02g263500*	*AT2G40280*	GO:0005576	extracellular region
*Glyma.02g263600*	*AT2G03430*	GO:0005886	plasma membrane
*Glyma.02g263900*	*AT5G45840*	GO:0005515	protein binding
*Glyma.02g265400*	Null	PF07172	glycine-rich protein family
*Glyma.02g265800*	Null	PF07172	glycine-rich protein family
*Glyma.02g270000*	*AT4G15410*	GO:0005515	protein binding
*Glyma.02g270700*	*AT3G21630*	GO:0005515	protein binding
*Glyma.02g271700*	*AT5G20930*	GO:0005515	protein binding
*Glyma.11g144900*	*AT5G45890*	GO:0005576	extracellular region
*Glyma.11g145800*	*AT5G60690*	GO:0003700	sequence-specific DNA binding transcription factor activity

Arabidopsis homologs were found in SoyBase (http://www.soybase.org, accessed on 16 May 2024); “Null” indicates no homologous genes were found in Arabidopsis.

## Data Availability

The datasets used and/or analyzed during the current study are available from the corresponding author upon reasonable request.

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
