# Peer review of "Densification of Genetic Map and Stable Quantitative Trait Locus Analysis for Amino Acid Content of Seed in Soybean (Glycine max L.)"

_plants, 2024, doi:10.3390/plants13152020_

Round 1

Reviewer 1 Report

Comments and Suggestions for Authors

The study by Xi et al., presents a comprehensive analysis of QTL aimed enhancing amino acid content in soybean. There are few comments and queries that need to be addressed before the acceptance of manuscript.

1. Explain why authors choose to map QTLs related to protein while few previous studies (cited below) already identified the QTLs associated with protein content. Explain the novelty of your work.

Park HR, Seo JH, Kang BK, Kim JH, Heo SV, Choi MS, Ko JY, Kim CS. QTLs and Candidate Genes for Seed Protein Content in Two Recombinant Inbred Line Populations of Soybean. Plants (Basel). 2023 Oct 16;12(20):3589. doi: 10.3390/plants12203589. PMID: 37896053; PMCID: PMC10610525.

Wang, J., Mao, L., Zeng, Z. et al. Genetic mapping high protein content QTL from soybean ‘Nanxiadou 25’ and candidate gene analysis. BMC Plant Biol 21, 388 (2021).

2. Did the authors identified any QTL associated to lipids along with protein, as oil and protein synthesis are highly interlinked metabolic pathways in soybean.

3. The reference to the results obtained in this study should be cited in the discussion section. Clearer labeling and references to figures within the discussion could help readers follow the results more easily.

4. Researchers predict candidate genes through ‘GO enrichment analysis’ but their functional validation is missing which could strengthen the findings.

5. Authors mention the impact of environment on quantitative traits, but it lacks detailed description on what and how they influence?

6. There are several grammar, typos and spelling mistakes.

line 15 ‘publicated’ should be replaced with ‘previously published’

line 30 ‘re considered’ remove ‘re’, it should be ‘considered as…..

line 158 and 159 ‘For the’ should not be used at the beginning of the sentence. Rephrase it

7. English editing is highly recommended.

8. Improve the labeling of the figure 4, it is not readable.

Comments on the Quality of English Language

English language editing is highly recommended.

Author Response

  1. Explain why authors choose to map QTLs related to protein while few previous studies (cited below) already identified the QTLs associated with protein content. Explain the novelty of your work.

Park HR, Seo JH, Kang BK, Kim JH, Heo SV, Choi MS, Ko JY, Kim CS. QTLs and Candidate Genes for Seed Protein Content in Two Recombinant Inbred Line Populations of Soybean. Plants (Basel). 2023 Oct 16;12(20):3589. doi: 10.3390/plants12203589. PMID: 37896053; PMCID: PMC10610525.

Wang, J., Mao, L., Zeng, Z. et al. Genetic mapping high protein content QTL from soybean ‘Nanxiadou 25’ and candidate gene analysis. BMC Plant Biol 21, 388 (2021).

Response 1: Thank you for pointing out this issue. In the work of Park HR and Wang, J, they identified several QTLs related to protein content in soybeans, which is highly valuable for enhancing soybean quality. However, soybean quality is constrained by numerous factors, among which amino acid content, particularly sulfur-containing amino acids such as methionine, is crucial. In this study, we primarily identified QTLs associated with essential amino acid content, providing genetic resources for improving soybean quality.

  1. Did the authors identified any QTL associated to lipids along with protein, as oil and protein synthesis are highly interlinked metabolic pathways in soybean.

Response 2: According to previous studies, we know that fat and protein content in soybeans are negatively correlated. the oil and protein content also showed this trend in our previous study. In addition, our laboratory has previously used this population for QTL mapping of oil content, the experimental results are not reflected in this article.

  1. The reference to the results obtained in this study should be cited in the discussion section. Clearer labeling and references to figures within the discussion could help readers follow the results more easily.

Response 3: We have added the Arabidopsis homology gene ID corresponding to the soybean gene in Table 4 to make the results clearer. At the same time, in the discussion section, we adjusted the content of the article to make its structure clearer.

  1. Researchers predict candidate genes through ‘GO enrichment analysis’ but their functional validation is missing which could strengthen the findings.

Response 4: We understand the functional validation of candidate genes may better strengthen the findings. However, The main purpose of this study is to quantify QTLs related to the content of essential amino acids, and although the current conclusions may not be the best, we believe that these candidate genes can be screened.

  1. Authors mention the impact of environment on quantitative traits, but it lacks detailed description on what and how they influence?

Response 5: This article is mainly concentrated on QTL. About the influence of different environments, we do found different QTLs in the three environments, which could be considered as the effect of the three environments. And the difference of QTL in different env is figured and shown on Table 2.

  1. There are several grammar, typos and spelling mistakes.

line 15 ‘publicated’ should be replaced with ‘previously published’

line 30 ‘re considered’ remove ‘re’, it should be ‘considered as…..

line 158 and 159 ‘For the’ should not be used at the beginning of the sentence. Rephrase it

Response 6: We have made the necessary corrections in the corresponding sections of the article and have thoroughly reviewed the entire text again. We greatly appreciate you pointing out these grammar, typos and spelling errors.

  1. English editing is highly recommended.

Response 7: Thank you for your valuable and thoughtful comments, we have checked and improve the English writing by an experienced English-speaking colleague in the revised manuscript.

  1. Improve the labeling of the figure 4, it is not readable.

Response 8: The labeling of the figure 4 has been corrected.

Reviewer 2 Report

Comments and Suggestions for Authors

This manuscript has useful information and is suitable for the purpose of Plants journal. 

However, there are grammatical errors and necessary to correct.

I attach revised paper.  

Comments on the Quality of English Language

There are grammatical errors and necessary to correct.

Author Response

This manuscript has useful information and is suitable for the purpose of Plants journal.

However, there are grammatical errors and necessary to correct.

Response : Thank you very much for your efforts in the review process. We have completed the revision according to your requirements in the corresponding position, and checked the possible errors in the article again.

Round 2

Reviewer 1 Report

Comments and Suggestions for Authors

1. Check the manuscript carefully, there are still typos and grammar errors

line 31 'value. [1,[2]. soybean' 

line 40 'particularly in the low content of sulfur-containing amino acids' replace 'with it is deficient in sulfur-containing amino acids'

line 68 'AS shown on Table 1' it should be 'As shown in Table 1'

2. Provide high quality figure 1. graph axis are not clear and unreadable.

3. Figure 2 legend write propability levels as '**=p<0.01, *=p<0.05'

4. Add the sources (data baases) used to extract Gene ID as footnote of Table 4.

5. Discussion section still need to be revised, I am unable to see any significant changes in the section. Moreover, figures and tables references are missing and results are not compared well with the previous findings.

6. Discussion section line 208-223 improve the datya presentation. there is no need to repeat 'in AT' again and again if you provide reference of Table 4  or you can rephrase the sentences for better presentation.

Comments on the Quality of English Language

Check the manuscript carefully, there are still typos and grammar errors

Author Response

  1. Check the manuscript carefully, there are still typos and grammar errors

line 31 'value. [1,[2]. soybean'

line 40 'particularly in the low content of sulfur-containing amino acids' replace 'with it is deficient in sulfur-containing amino acids'

line 68 'AS shown on Table 1' it should be 'As shown in Table 1'

Response 1: Thank you very much for your careful guidance and help, we have corrected the wrong part of the article according to your requirements, and have checked the grammar and spelling errors in the whole article again.

  1. Provide high quality figure 1. graph axis are not clear and unreadable.

Response 2: Thank you for pointing out the problem, I have reinserted the high resolution picture.

  1. Figure 2 legend write propability levels as '**=p<0.01, *=p<0.05'

Response 3: The description of the significance level in Figure 2 has been changed in accordance with your request, thank you for your valuable comments.

  1. Add the sources (data bases) used to extract Gene ID as footnote of Table 4.

Response 4: Thank you very much for pointing out this problem, adding the sources used to extract Gene ID as footnote of Table 4 can help make the data source in the article more clear, we have modified according to your request.

  1. Discussion section still need to be revised, I am unable to see any significant changes in the section. Moreover, figures and tables references are missing and results are not compared well with the previous findings.

Response 5: We have rewritten the discussion part and added some comparison with previous studies, but due to the short time, there may still be some problems in our content. Thank you for your understanding and help!

  1. Discussion section line 208-223 improve the datya presentation. there is no need to repeat 'in AT' again and again if you provide reference of Table 4 or you can rephrase the sentences for better presentation.

Response 6: In the discussion part, we have been repeatedly using "in AT", which seems to be repetitive and redundant in the expression of the paragraph. By adding the content of Table 4, we can indeed omit this part. Thank you for your suggestion, which is of great help to improve the content of our article.
